# Zebrafish Larvae Behavior Models as a Tool for Drug Screenings and Pre-Clinical Trials: A Review

**DOI:** 10.3390/ijms23126647

**Published:** 2022-06-14

**Authors:** João Gabriel Santos Rosa, Carla Lima, Monica Lopes-Ferreira

**Affiliations:** Immunoregulation Unit of the Laboratory of Applied Toxinology (CeTICs/FAPESP), Butantan Institute, São Paulo 05503-900, Brazil; joao.rosa@esib.butantan.gov.br (J.G.S.R.); carla.lima@butantan.gov.br (C.L.)

**Keywords:** embryo-larval stage, alternative model, 3Rs, behavioral repertoire, drug discovery, neurological drugs, neurotransmitters, anxiety-like behavior

## Abstract

To discover new molecules or review the biological activity and toxicity of therapeutic substances, drug development, and research relies on robust biological systems to obtain reliable results. Phenotype-based screenings can transpose the organism’s compensatory pathways by adopting multi-target strategies for treating complex diseases, and zebrafish emerged as an important model for biomedical research and drug screenings. Zebrafish’s clear correlation between neuro-anatomical and physiological features and behavior is very similar to that verified in mammals, enabling the construction of reliable and relevant experimental models for neurological disorders research. Zebrafish presents highly conserved physiological pathways that are found in higher vertebrates, including mammals, along with a robust behavioral repertoire. Moreover, it is very sensitive to pharmacological/environmental manipulations, and these behavioral phenotypes are detected in both larvae and adults. These advantages align with the 3Rs concept and qualify the zebrafish as a powerful tool for drug screenings and pre-clinical trials. This review highlights important behavioral domains studied in zebrafish larvae and their neurotransmitter systems and summarizes currently used techniques to evaluate and quantify zebrafish larvae behavior in laboratory studies.

## 1. Zebrafish as a Model for Phenotype-Based Screening

An analysis of first-in-class drugs approved by the U.S. Food and Drug Administration (FDA) between 1999 and 2008 revealed that 62% of them were discovered by phenotype-based screening [1], in which the assay results in an organic/functional phenotype that integrates multiple biochemical signals from the biological system. Phenotype-based screening combines general morphological screening for abnormalities in embryonic development with observations of animal behavior due to observation of an intact animal and the interaction between different cell types. Thus, during the process, the identified hits have higher biomedical applicability [2].

Several factors may explain the apparent superiority of phenotype-based screening over target-based screening, which starts from a specific molecular target hypothesized to play a key role in disease. First, phenotypic screening can discover effective drugs in the absence of a validated target. Second, it can identify compounds that produce a therapeutic effect through simultaneous activity on multiple targets. Third, phenotypic screening often combines screening and counter-screening in the same assay, finding compounds that produce the desired effect while analyzing undesirable qualities. As a direct result, compounds that advance from phenotype-based screening are generally safer, more effective, and with fewer side effects when compared to those obtained from in vitro cell culture tests using target-based screenings [3].

Zebrafish (*Danio rerio*) is a teleost fish that has highly conserved molecular targets and physiological pathways among vertebrates, with approximately 70% of homology to human genes [4], and the neuroanatomical features, neuronal cells morphology, and circuits are similar to mammals [5]. Screening for new molecules using zebrafish represents a small but growing fraction of phenotype-based screening (Figure 1). In addition to discovering novel compounds with therapeutic potential, zebrafish screenings have proven useful for identifying novel uses for existing drugs. Zebrafish screenings provide, in addition to the typical advantages of phenotype-based screenings, the unique advantage of being performed on a vertebrate organism, embryo, or larva. Pain, sedation, tumor metastasis, vascular tone, and intestinal motility are some relevant examples of phenotypes that are observable in zebrafish yet simply inaccessible in cell culture.

Drug screenings using the zebrafish embryo-larval stage fit the concept of replacement, reduction, and refinement (3Rs) in an effective way and can replace other vertebrates in studies involving all organic systems [6]. Zebrafish independent feeding comprises several criteria such as a completely developed digestive tract, swimming ability to pursue the prey/food, and total yolk depletion, and these physiological events occur between 120 and 144 hpf [7]. Concerning behavioral features, zebrafish are able to exhibit coordinate behavior in response to different stimuli and present major neuromodulatory systems by 96 hpf [8,9]. Thus, according to Strahle et al. (2012), larvae below 120 hpf are considered an alternative model to animal research once they are classified as non-protected under the EU Directive of animal welfare (EU 2010/63/EU) [6]. However, research using early-life stages of zebrafish still obeys the concept of 3Rs [10].

The advantages of using an intact animal as a focus for screening are particularly evident for neurologic drug discovery, where the complexities of cell-cell interactions and endocrine signaling are challenging. In addition, behavior analysis can be combined with transgenic strains using methods to generate targeted genetic modification such as Clustered Regularly Interspaced Short Palindromic Repeats-associate protein 9 (CRISPR/Cas9), RNA interference (RNAi), zinc-finger nucleases (ZFNs), and antisense oligonucleotide morpholinos [11].

Whereas cell-based assays provide limited information on the absorption, distribution, metabolism, excretion, and toxicity (ADME-Tox) of compounds, zebrafish screenings reveal insights into these pharmacological characteristics as zebrafish larvae have functional livers, kidneys, and blood–brain (BBB) and blood–retinal barriers (BRB) [12,13], as well as drug-metabolizing enzymes and metabolic rates comparable to humans [14]. To produce phenotypes in vivo in zebrafish assays, compounds must exhibit the ability to be absorbed, reach target tissue, and avoid rapid metabolism and excretion. This fact may explain the observation that several compounds that were discovered in zebrafish screenings were rapidly translated to mammalian models in vivo with minimal optimization of pharmacological properties.

Zebrafish shares extensive homologies to other vertebrate species (including rodents and humans) in terms of brain patterning and the structure and function of several neural and physiological systems, including the stress-regulating axis [15]. The formation of the zebrafish neurological system in the larval stage occurs between 72 hpf and 120 hpf. During development, three embryonic layers (endoderm, mesoderm, and ectoderm) are generated, from which neural tissue originates. The anterior neural tube undergoes a series of curves and constrictions to subdivide into the forebrain, midbrain, and hindbrain, allowing it to settle into the skull. The first morphogenetic event is the formation of a constriction at the midbrain-hindbrain boundary. Another key event in brain morphogenesis is the opening of the cerebral ventricles [4,16]. Zebrafish primary neurons appear by 24 hpf, and few axonal tracts and commissures are observed in early embryogenesis. The morphological development of the CNS is completed with the end of embryogenesis (≈72 hpf), with the BBB present at an earlier stage [17]. In contrast to mammals, the zebrafish brain has an extraordinary ability to regenerate [18], which represents another advantageous feature in exploring the mechanisms underlying neuroprotection, neurogenesis, and functional integration of newborn neural cells and the search for new drugs for neurodegenerative or neurodevelopmental disorders.

Chemical screenings rely on biological models, generally well characterized and suitable for extrapolation to other vertebrates. Ideally, the use of models with established methods is required for reliable results. Zebrafish’s behavioral repertoire in the larval stage is limited since the nervous system is still in development, and several neurocircuits could be absent or underdeveloped. Thus, besides being an undoubtedly suitable biomedical model, zebrafish larvae may not express certain complex behavior. Behaviors such as schooling, aggressive encounters, and mating only appear during the transition to adults, with these behaviors comprising social behavior domains, which become fully developed in adult fishes [9,19].

Zebrafish have been used when it is desired to correlate behavioral patterns, such as cognition, social interactions, and locomotor activity, with physiological evidence of a specific disorder. Different methods and disease-relevant models, which are capable of producing reliable information for behavioral pharmacology, have been developed and adapted for zebrafish to use in neurological drug screening during the last decade [20,21,22,23]. The effects of different new compounds on brain development can be assessed by different neurobehavioral parameters, including swimming capacity and functionality of the motor, sensory, and stress-regulating systems.

## 2. Zebrafish Neurotransmitter Systems

The neuromodulatory circuits present in mammals can also be found in the larval zebrafish brain. Major neurotransmitter systems are conserved through vertebrates, and in zebrafish, glutamate, gamma-aminobutyric acid (GABA), acetylcholine, dopamine, serotonin (hydroxytryptamine or 5-HT), noradrenaline, and histamine systems are well described [24]. Alterations in patterns of transmission are related to neurological disorders [25].

The main excitatory neurotransmitter in vertebrates is glutamate, which regulates synaptic transmission and neuronal excitability. The 24 hpf zebrafish larvae present vesicular glutamate transporters (VGLUT2), and by 96 hpf zebrafish larvae express VGLUT 1 and VGLUT 2, besides glutamate metabotropic and ionotropic receptors in the olfactory bulb, optic tectum, hypothalamus, cerebellum, and retina [26]. GABA is an inhibitory neurotransmitter expressed both in early-life stages and in adulthood [27,28]. In zebrafish, GABA-ergic neurons appear in the olfactory bulb, subpallium, posterior preoptic area, the diencephalic basal plate, the central optic tectum, torus semicircularis, ventral mesencephalic tegmentum, valvula of the cerebellum and medulla oblongata [29].

Zebrafish also present catecholamines as a major neurotransmitter. Noradrenaline (NA) acts in the autonomic nervous system and controls cognition, including learning and memory, as well as arousal and reward systems; the zebrafish noradrenergic system is very similar to mammals [30,31]. In the same way, the histaminergic system is quite similar to mammalian and exerts an effect on memory, cognition, and circadian rhythm [32]. Serotonin (5-HT) is a neurotransmitter present in the embryonic stage in the spinal cord and in the telencephalon, hindbrain, and the raphe region in zebrafish larvae and adults, with a clear correlation between life stages [31].

Behavioral processes such as aggression, anxiety, cognition, and sleep are modulated by 5-HT. According to Ek et al. (2016), zebrafish have all the dopamine receptors, except dopamine receptor type 5 D5 [33]. The dopaminergic system is an important player in the regulation of locomotion of zebrafish larvae. Furthermore, zebrafish present correlated behavioral phenotypes to rats and humans, followed by dopaminergic system manipulation [34,35].

Under stress conditions, zebrafish activate the hypothalamus-pituitary-interrenal (HPI) neuroendocrine axis, culminating in cortisol secretion, similar to humans. Cortisol binds to glucocorticoid receptors, regulating transcriptional responses related to glucose metabolism, ion regulation, immune system, and, ultimately, behavior [36]. Cholinergic neurons appear at the embryonic stage and are amply distributed in the CNS of adults. The enzyme acetylcholinesterase (AChE) expression is initially found in 4 hpf embryos and increases by 210-folds in 144 hpf larvae [37]. In zebrafish, similarly to humans, acetylcholine acts in cholinergic receptors, muscarinic and nicotinic [38,39], modulating cognitive processes.

The similarity in the zebrafish neuroendocrine repertoire added to easy genetic and pharmacological modulation allows the reproduction of complex behavioral models that mirror those of human neurological disorders such as Alzheimer’s disease [40,41], Parkinson’s disease [42,43], depression and anxiety [28,44], epilepsy [45,46,47], and amyotrophic lateral sclerosis—ALS [48,49]. In this way, the use of behavioral models in zebrafish larvae supports the study and development of drugs for the CNS.

## 3. Neurological Functions and Behavior Models toward Pre-Clinical Assays

Zebrafish are diurnal and can accomplish behavioral tasks under a normal light setting. Tests with zebrafish can be performed quickly, with large numbers of compounds in parallel in contained testing arenas, through the integration of infrared cameras with programmable stimuli control. Moreover, in experimental conditions, all behavioral phenomena can be quantified by high-level automated tools [50,51]. In addition to generating results as satisfactory as those obtained in experiments with adults, it provides a reduction in the time of experiments and in the size of the apparatus for carrying them out, which makes studies with zebrafish larvae more efficient.

Neurological functions such as spatial and social learning, memory, anxiety, and social or sickness behaviors driven by neurotransmitters can be exploited in zebrafish larvae behavior models. Behavioral phenotypes detected in both larval and adult stages of zebrafish can be separated by social behavior (exploratory and locomotor abilities) and sickness behavior, which is characterized by lethargy, anxiety, reduced physiological function such as locomotor activity, and exploratory and social interaction [52]. Escape swimming in response to touch and sound is a reflexive response, despite great complexity.

Behavioral phenomena such as memory or processing and spatial learning take place in the lateral pallium of the telencephalic area of zebrafish, and fear response is associated with the habenula, like mammalian hippocampus and amygdala, respectively [53,54,55]. Among the main types of learning and memory models in zebrafish larvae, it is possible to mention habituation, characterized by an animal’s response to repeated stimuli; sensitization, based on painful or noxious stimuli; conditioning, which consists of associating a neutral stimulus with a reinforcing stimulus; and social learning, based on the animal’s preference to shoal formation [30].

Furthermore, zebrafish explore novel objects or environments with more emphasis than the known ones [56,57], and according to Santacà, Dadda, Petrazzini, and Bisazza (2021), zebrafish can distinguish novel or known objects with different sizes, shapes, and colors [58]. Besides this visual discrimination learning, zebrafish show a robust cognitive repertoire such as avoidance learning, spatial learning, and reinforcement-based learning [59]. Environmental novelty induces a robust behavioral response in zebrafish, in both larvae and adults [55,60,61]. As in other vertebrates, the zebrafish response to novelty is consistent with an anxiety-like behavior; since the same neurotransmitters and neuroendocrine system are also present [29,62,63].

In a stressful situation, which evokes the activation of the HPI axis and the action of the corticotropin-releasing hormone (CRH) cascade, culminating with cortisol release, zebrafish response is consistent with an anxiety-like behavior [60,61,64]. In behavioral assays, anxiety states are reflected by reduced exploration, and this behavior is clearly demonstrated by zebrafish, along with freezing episodes and erratic movement [65,66,67]. Next, we summarize contemporary outlooks on methods applied to evaluate larval zebrafish behavior and discuss its applications in toxicological pre-clinical screening for drug discovery (Table 1).

### 3.1. Anxiety-like Behavioral Models

Locomotor activity begins in the early stages, and at 17 hpf, first movements appear as spontaneous contractions inside the chorion, and in the next hours of development, the movements become more coordinated, and the larvae are able to respond to physical contact at 21 hpf. The 24–48 hpf zebrafish larvae present coordinated swimming, as the nervous system continues to develop, and at 96–120 hpf show brain high complexity, expressing more robust behaviors [8]. To assess locomotor capability, larvae are placed individually in a plate or multi-well plate and acclimated to the experimental setup. Then, all movements are tracked, and parameters of total distance traveled and mean velocity are analyzed to assess potential neurobehavioral effects of the treatment [68,69]. Zebrafish hyper- or hypoactivity is a valuable biological marker of environmental alterations—natural or in a laboratory—since zebrafish respond in a fast manner by altering locomotor parameters such as distance moved and mean velocity, which could lead to changes in exploration pattern [70].

Afrikanova et al. (2013) validated the model of zebrafish larval (144 hpf) locomotor assay as a rapid first-pass screening tool for assessing the anticonvulsant and/or proconvulsant activity of compounds [71]. They used the automated tracking device ZebraBox™ [84] to evaluate acute locomotor impairment in zebrafish larvae, as well as their escape response upon slightly disturbing the tail with a fine needle.

Visual motor response (VMR) assay is a startle response-like test mediated by vision and started by a drastic change in light condition designed to evaluate both locomotion ability and visual capacity of zebrafish larvae [72,73], and larvae between 120 and 168 hpf produce more robust responses [74,75,85]. In this test, zebrafish larvae are exposed to alternating light and dark periods, and individuals with a functional visual system should respond with specific swimming patterns. The larvae movement in both dark and light periods is recorded and quantified in a high-throughput manner generating reliable data [86,87] through automated systems such as ZebraBox™ apparatus [84] or DanioVision™ [88]. VMR is a useful tool to assess oculotoxicity and visual disorders, as well as to verify the physiological integrity of neurocircuits promoted by therapeutic compounds in a noninvasively way [83,89,90,91].

Monoamines perform a significant role in the neurocircuits in vertebrates, including zebrafish [92,93,94], and pharmacological manipulation by anxiolytic drugs and inhibitors (MAOI) supports its involvement in locomotor activity. Cunha et al. (2018) evaluated the effect of fluoxetine in 1 hpf embryos by evaluating the transcription of genes involved in serotonin, dopamine, and adrenergic transporters and receptors signaling and by evaluating the escape behavior after touch in 80 hpf larvae [95]. Faria et al. (2021) reported that the inhibition of monoamine oxidase (MAO) activity by deprenyl—an MAOB inhibitor—resulted in enhanced levels of serotonin and dopamine characteristic of anxiolytic-like behavior [76]. Exposed larvae present a reduced response in behavioral paradigms evaluated by visual motor response (VMR).

Recently, a series of behavioral tests such as anxiety, inattentive behavior, and circling behavior in valproic acid-treated larvae (168 hpf) were performed as surrogate parameters of autism spectrum disorder (ASD)-like characteristics [77] for quick screening of standard drugs marketed for symptomatic treatment in ASD. In conclusion, the authors propose the use of 7-day-old larval zebrafish for preliminary screening of drugs against VPA-induced neurodevelopmental toxicity [96].

Locomotor parameters such as total distance moved, swim speed, erratic movements, burst swimming, or freezing are indicatives of locomotor capacity and/or anxiety-like state of zebrafish [97]. In behavioral light/dark preference tests without aversive stimulus, non-anxious larvae tend to increase exploratory activity by spending time in the center zone or in the bright zones (phototaxis) [98]. Chen, Deb, Bahl, and Engert (2021) consider that zebrafish larvae present a phototactic behavior aiming to manage the environmental luminance to a level suitable to their visual function, escaping from dark areas. Thus, zebrafish larvae with decreased anxiety levels exhibit positive phototaxis [99].

Thompson et al. (2017) reported that zebrafish larvae exposed to serotonin-norepinephrine-dopamine reuptake inhibitor (SNDRI) venlafaxine were less active and covered shorter distances in a light/dark behavioral test compared to the controls [78]. In addition, other studies have shown that in zebrafish larvae as young as 120 hpf exposed to diazepam, the thigmotaxis was significantly attenuated, confirming the relation of the GABAergic signaling pathway in anxiety behavior [79]. Han et al. (2021), through thigmotaxis behavior, described a non-anxiogenic activity of neuroserpine [80]. In addition, Maphanga et al. (2022) used thigmotaxis behavior in zebrafish larvae to confirm the anxiolytic-like effect of *Mesembryanthemum tortuosum* L. extract, known for presenting the capacity to alleviate anxiety, stress, and depression [81].

Copmans et al. (2019) identified two known isoquinoline alkaloids, TMC-120A and TMC-120B, as novel antiseizure compounds, which were tested using the zebrafish larvae photomotor response (PMR) assay [82]. The PMR is a stereotypical behavior of 30–40 hpf zebrafish embryos, observed by the embryo’s movements inside the chorion, and is triggered by high-intensity light [21,100].

### 3.2. Visual Behavior

Zebrafish larvae are a well-suited model to investigate visual behavior, in other words, alteration of locomotor parameters mediated by visual stimuli. The optomotor response (OMR) and optokinetic response (OKR) are innate visual responses in which an individual moves the eye (OKR) and/or swims in the direction of an optic flow (OMR) and mimics a naturalistic behavior in larval zebrafish, aiming to stabilize the image around [8]. Another visual behavior, the visuomotor response (VMR), aims to evaluate both locomotion ability and visual capacity of zebrafish larvae. In this test, zebrafish larvae are exposed to alternating light and dark periods, and individuals with a functional visual system should respond by showing specific swimming patterns [8].

Zebrafish characteristic of optical transparency of early-life stages allows the observation of the functioning of visuo-neural circuitry, and secondly, the pattern of behavior can be measured and evaluated [90]. Comparable to the human eye, the mature zebrafish retina is composed of three nuclear layers separated by outer and inner plexiform layers (OPL and IPL, respectively). Photoreceptor (rod and cones) cell bodies reside in the outer nuclear layer (ONL); the amacrine, bipolar, horizontal, and Müller glial cell bodies occupy the inner nuclear layer (INL), and the ganglion cell bodies are contained in the ganglion cell layer (GCL). Synapsis between these nuclear layers and retinal neurons occurs at the plexiform layers [101].

In phenotypes-based screenings, the visual system is a powerful tool to assess drug effects since it presents a rapid development [102]. Robust visually guided responses such as VMR can be used to identify new therapeutic molecules or to reposition already approved drugs [83].

Several neurotransmitters coordinate the visual function in zebrafish. The photoreceptor layer communicates to bipolar cells via the glutamate pathway, and Lovett-Barron et al. (2017) described monoaminergic and cholinergic systems acting on modulation of visual sensory information [103]. Furthermore, the dopaminergic system in the posterior tuberculum and serotonergic system in the dorsal raphe nucleus are implicated in visuomotor response and short-term memory to diverse stimuli [104,105,106].

## 4. Concluding Remarks

As summarized in this review, there are multiple forms to apply the zebrafish in behavioral research. As an alternative model, zebrafish larvae present many features that qualify as a suitable model for translational biomedical research. Neurocircuits responsible for behavior in zebrafish are physiologically and anatomically similar to mammals, as well as the mechanisms involved in drug responses. Behavioral responses in zebrafish larvae, such as changes in movement and locomotion profile, driven by complex neural circuits that include perception, cognition, and decision-making processes, and visuomotor functions can be reduced or exacerbated by different stimuli and explored by the use of several tests and devices specifically designed for zebrafish. The modulated behavioral repertoire can be used for the screening and discovery of candidate neurological drugs in the pre-clinical stage of development and clinically translated for various neurological diseases (Figure 2).

## Figures and Tables

**Figure 1 ijms-23-06647-f001:**
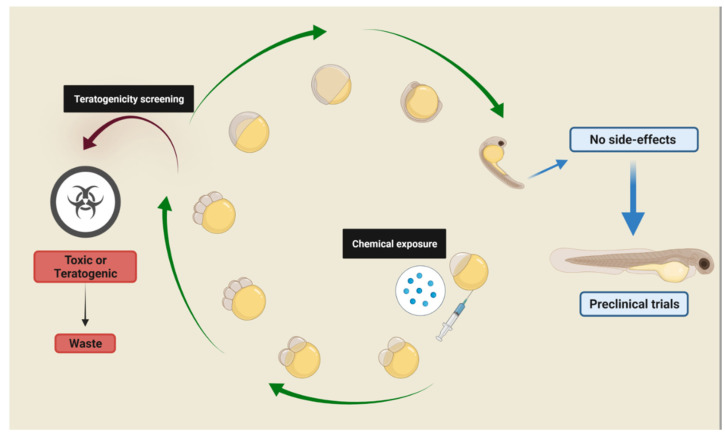
Zebrafish development. Zebrafish embryo-larval stages permit the phenotypic observation of the whole organism in a large-scale chemical screening. If a molecule or drug is unable to induce death or teratogenicity during the period of embryonic development, the larvae can be used to assess changes in behavior.

**Figure 2 ijms-23-06647-f002:**
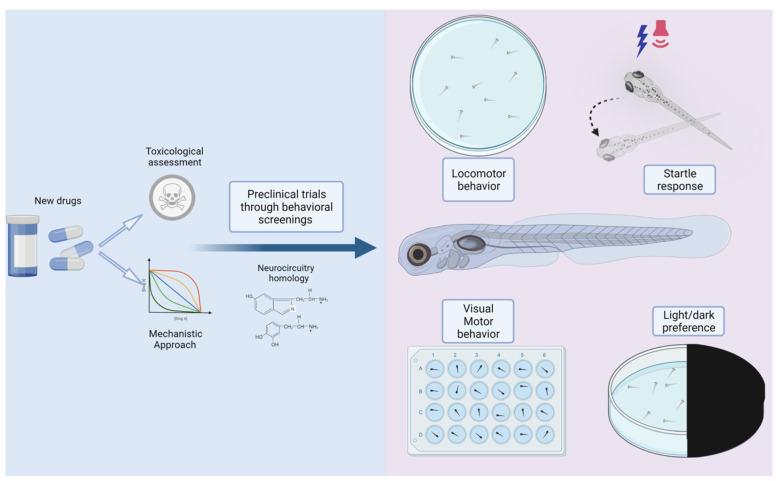
Behavioral response in zebrafish larvae. Changes in movement and locomotion profile, driven by complex neural circuits that include perception, cognition and decision-make processes, and visuomotor functions, can be used for the screening and discovery of candidate neurological drugs in the pre-clinical stage of development.

**Table 1 ijms-23-06647-t001:** Selected studies using zebrafish larvae as experimental model in behavior paradigms.

BEHAVIORAL TEST	ENDPOINTS	REFERENCE
LIGHT-DARK TEST	Total distance traveled	[68]
VISUAL MOTOR RESPONSE	Velocity, total distance moved, and mobility time	[69]
LOCOMOTOR ACTIVITY	Velocity, total distance moved, and mobility time	[70]
LOCOMOTOR ACTIVITY	Total distance traveled	[71]
ACOUSTIC STARTLE RESPONSE	Head angle	[72]
VISUAL MOTOR RESPONSE	Total distance traveled	[73]
VISUAL MOTOR RESPONSE	Average distance traveled	[74]
VISUAL MOTOR RESPONSE	Burst swim	[75]
VISUAL MOTOR RESPONSE	Total distance traveled	[76]
VIBRATIONAL STARTLE RESPONSE	Total distance traveled	[76]
LOCOMOTOR ACTIVITY	Total distance traveled, mean speed, turn angle	[77]
THIGMOTAXIS	Entries in outer area	[77]
LIGHT-DARK TEST	Total distance traveled	[78]
THIGMOTAXIS	Distance traveled in outer area	[78]
THIGMOTAXIS	Percentage of distance moved in outer zone	[79]
VISUAL MOTOR RESPONSE	Total distance traveled	[80]
THIGMOTAXIS	Distance traveled/time spent in each zone	[80]
THIGMOTAXIS	Percentage of distance moved in outer zone	[81]
LOCOMOTOR ACTIVITY	Average distance traveled	[81]
PHOTOMOTOR RESPONSE	Movements/5 min	[82]
LOCOMOTOR ACTIVITY	Total distance traveled	[82]
VISUAL MOTOR RESPONSE	Total distance traveled	[83]

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
