# Peer review of "Zebrafish Larvae Behavior Models as a Tool for Drug Screenings and Pre-Clinical Trials: A Review"

_ijms, 2022, doi:10.3390/ijms23126647_

Round 1
Reviewer 1 Report
Zebrafish (Danio rerio) is a very popular study species, and zebrafish larvae represent a convenient and powerful model to study the bioactivity and toxicity of drugs. Here, the authors review the advantages of phenotype-based screening with zebrafish for drug characterization, primarily focusing on behavioral assays with larvae. In addition, they also collated some helpful information on the various neurotransmitter systems in zebrafish and their relevance for behavioral testing in the context of drug characterization.
Although this review does not per se provide any novel perspective, it provides a solid background on the use of zebrafish for drug screenings and, from that perspective, is a useful paper for people that are new to the field and want to get familiar with the subject.
My largest point of criticism is related to the fact that the authors only discussed the advantages of zebrafish larvae behavioral testing, but did not discuss its potential limitations (and the way forward, when relevant). In order to provide a complete and non-biased overview of the state-of-the-art, I believe it is critical to also discuss such limitations (which does not diminish the usefulness of zebrafish larval tests). For instance, even though I agree that zebrafish larvae are useful and practical models for behavioral research, it should be recognized that their behavioral repertoire is much more rudimentary compared to adult fish (e.g. no spawning behavior, no shoaling, no ageing-related pathologies) which may limit a full characterization of the bioactivity and toxicity of, for instance, endocrine and neuroactive drugs. For this reason, behavioral tests with fish in later life-stages (despite being classified as laboratory animals) are on the rise again, which I believe is something that should be recognized. Another limitation is, for instance, that the natural history of zebrafish is still poorly known, which hampers the welfare of fish and the quality of scientific research (see DOI: 10.1111/brv.12831). Discussing such points would definitely provide a more complete and balanced overview on the matter, and would further improve the paper.
Please find my line-by-line comments below, which I hope will help you to further fine-tune your manuscript. I provided some reference to literature to support my claims and to help you in addressing the comments. I hope you will find my review constructive and helpful, and please be assured that any negative tone that may emerge between the lines is entirely unintentional.
Great work so far, and good luck!
ABSTRACT
L12: “to review currently therapeutic substances” is a bit vague, I suggest to make this more concrete. Perhaps this: “To review the biological activity and toxicity of therapeutic substances”?
L13: Please change “in trustful biological systems to obtain effective results” to something along the lines of “on robust biological systems to obtain reliable results”.
L13-14: I’m not really sure if I understand what you mean with the first part of this sentence. Can you please rephrase to clarify?
L18: You mean to say that various physiological pathways that are important in humans are conserved in zebrafish, but the specific phrasing “a highly conserved genome” is a bit odd – please rephrase.
L19: Please change “repertory” to “repertoire”.
Keywords: Note that keywords that overlap with the title are superfluous (e.g. zebrafish, pre-clinical trials). I suggest to choose other keywords to replace the overlapping ones, it will increase the visibility of your paper.
1. Zebrafish as a Model for Phenotype-Based Screening
L31-33: “that integrates multiple biochemical signals from the native biological context” This is rather vague, can you please rephrase to clarify?
L35: “multiple signaling mechanisms” is vague, please phrase this in more concrete terms to clarify. Do you mean to say that “phenotype-based screening integrates physiological, morphological and behavioral data”?
L38-39: “which verify a specific molecular target in a heterologous context” is not very clear. Do you mean “which starts from a specific molecular target hypothesized to play a key role in disease”?
L44: So compounds that “are generally of better quality” are compounds with less side-effects, right? Or what exactly do you mean with the “quality” of a compound? (Because also things like efficacy, potency, etc. could be considered hallmarks of quality.)
L46: See my earlier comment about “highly conserved genome”. Molecular targets and physiological pathways and so on can be evolutionarily conserved, but a “conserved genome” is a bit odd (so I suggest to rephrase a bit).
L48: It would be interesting if you could briefly summarize here what are the most commonly used phenotype-based screenings, if not with zebrafish.
L56-67: I agree that working with embryos is a way to accommodate legislation regarding the ethical usage of animals, and I also agree that extrapolations can be made from zebrafish embryos to other vertebrates (including humans). Still, this is a double-faced coin, which means that there are not only advantages but also limitations to be considered. One important limitation is that larvae have a very rudimentary behavioral repertoire compared to adult fish, and are in that sense less versatile. For instance, larvae do not engage in mating/spawning behavior or full-extent shoaling behavior, but these can nevertheless be very relevant traits. As another example, testing the bioactivity of compounds to treat ageing/senescence-related diseases requires the use of adult/senescent fish rather than embryos or larvae. To characterize the bioactivity and toxicity of compounds to a fuller extent, recent papers in fish behavioral research have advocated for the use of fish in later life-stages in addition to only embryos or larvae, and I fully agree with that development (see e.g. DOI: 10.1002/etc.4301 and DOI: 10.1016/j.chemosphere.2021.129697). I believe some discussion on this point should be added to the manuscript (it can be later on), to recognize that there are also limitations to embryo/larvae-based fish testing.
L63-67: Research with zebrafish early-life stages “obeys the concept of 3Rs” from the sense that they are not considered as laboratory animals (“Replacement”) and are therefore not subjected to any legislation regarding the ethical use of laboratory animals. It’s basically a legal backdoor. Building on my previous comment, I would add that tests with adult (zebra)fish can also be conform the legal ethical framework, by investing in “Reduction” and “Refinement”, as discussed in DOI: 10.1111/jfb.14512 and DOI: 10.1016/j.chemosphere.2021.129697 by the aforementioned author-group. In my opinion, adding such perspective is important because it gives a more nuanced and complete picture of the latest developments in the field of fish testing.
L75-84: I fully agree, great section!
2. Zebrafish Neurotransmitter Systems
L126-128: I suggest to tweak this sentence a bit, for instance like: “ Noradrenaline (NA) acts on the autonomic nervous system and controls cognition including learning and memory, as well as arousal and reward systems.”
L143: Please explain the abbreviation upon first encounter (please also check this for other abbreviations, I didn’t verify it all).
L148-153: Yes, but see my above comments on the limitations of larvae tests
3. Neurological Functions and Behavior Models Towards Pre-Clinical Assays
L159-161: I agree, tests with zebrafish larvae are much more efficient compared to when you use older animals. But again, I believe that some discussion of its limitations is necessary to provide a balanced view.
L179-187: Please specify if these statements are specifically for adults, larvae, or both.
L192-194: I like that you provide an overview of which behavioral assays exist in zebrafish, great work! Are these the only established zebrafish larvae behavioral tests, or are there others (i.e. is this list exhaustive)?
L206-207: “since zebrafish responds in a fast manner by altering another behavior like exploration and boldness” This part of the sentence, and its connection with activity level is not so clear. Please rephrase to clarify.
L210-213: This sentence needs some tweaking. Perhaps something like this? “They used the automated tracking device ZebraBoxTM to evaluate acute locomotor impairment in zebrafish larvae, as well as their escape response upon slightly disturbing the tail with a fine needle.”
L225: Please change “performs” to “perform”.
L241-245: This explanation is a bit confusing, please clarify (i.e. increased anxiety is reflected by reduced activity and by spending less time in the open and illuminated area).
L245-248: I’m not sure if I get your point here, could you please rephrase to clarify?
L264: I must admit that this paragraph confuses me a bit, primarily because I’m not entirely sure what you mean by “visual behavior”? Do you mean the movement of the eyes? How does one make these observations (i.e. which equipment/assays etc.)?
4. Concluding remarks
L287: Please change “decision-make processes” to “decision-making processes”. (Also applies for L308).
L289: Please change “apparatus” to “apparatuses” or “devices”.
Figure 1: The figure is visually very attractive, great work! One point for improvement, however, is that the link between the displayed behaviors (purple box) and the text is not so clear. In the text, you discussed 4 assays, in the following order: 1. locomotor test in a Petri-plate or multi-well plate, 2. escape test upon disturbance, 3. visual motor response, and 4. light/dark preference test. The order in the figure, as well as the name of the tests, seems to be different. Please use the same order and name for clarity/consistency.
Author Response
Manuscript ID: ijms-1754880
Type of manuscript: Review
Title: Zebrafish larvae behavior models as a tool for drug screenings and pre-clinical trials: a review
Authors: Joao Gabriel Santos Rosa, Carla Lima, Monica Lopes-Ferreira
Reviewer #1
First of all, we appreciate you taking the time out to share your comments with us. We value and respect your opinion. We agree with the need to make the text more appropriate for the Journal's audience, and we have already adjusted the manuscript, including additional changes as suggested.
ABSTRACT
L12: “to review currently therapeutic substances” is a bit vague, I suggest to make this more concrete. Perhaps this: “To review the biological activity and toxicity of therapeutic substances”?
Answer: the phrase was rewritten in another way, keeping the original meaning: “review the biological activity and toxicity of therapeutic substances,” as suggested.
L13: Please change “in trustful biological systems to obtain effective results” to something along the lines of “on robust biological systems to obtain reliable results”.
Answer: the phrase was rewritten in another way, keeping the original meaning: “research relies on robust biological systems to obtain reliable results.”, as suggested.
L13-14: I’m not really sure if I understand what you mean with the first part of this sentence. Can you please rephrase to clarify?
Answer: the phrase was rewritten in another way, keeping the original meaning: “Phenotype-based screenings can transpose the organism's compensatory pathways by adopting multi-targeted strategies for the treatment of complex diseases,…
L18: You mean to say that various physiological pathways that are important in humans are conserved in zebrafish, but the specific phrasing “a highly conserved genome” is a bit odd – please rephrase.
Answer: the phrase was rewritten in another way, keeping the original meaning: “conserved physiological pathways that are found in higher vertebrates, including mammals, along with….”, as suggested.
L19: Please change “repertory” to “repertoire”.
Answer: The word was altered to “repertoire”, as suggested.
Keywords: Note that keywords that overlap with the title are superfluous (e.g. zebrafish, pre-clinical trials). I suggest to choose other keywords to replace the overlapping ones, it will increase the visibility of your paper.
Answer: The follow keywords were suppressed: “Zebrafish; Behavior models; pre-clinical assays”, and the words “3Rs; neurotransmitters, anxiety-like behavior” were added.
- Zebrafish as a Model for Phenotype-Based Screening
L31-33: “that integrates multiple biochemical signals from the native biological context” This is rather vague, can you please rephrase to clarify?
Answer: the phrase was rewritten in another way, keeping the original meaning: “… which the assay results is an organic/functional phenotype that integrates multiple biochemical signals from the biological system.”
L35: “multiple signaling mechanisms” is vague, please phrase this in more concrete terms to clarify. Do you mean to say that “phenotype-based screening integrates physiological, morphological and behavioral data”?
Answer: the phrase was rewritten in another way, keeping the original meaning: “As phenotype-based screening integrates physiological, morphological and behavioral data …”, as suggested.
L38-39: “which verify a specific molecular target in a heterologous context” is not very clear. Do you mean “which starts from a specific molecular target hypothesized to play a key role in disease”?
Answer: the phrase was rewritten as suggested.
L44: So compounds that “are generally of better quality” are compounds with less side-effects, right? Or what exactly do you mean with the “quality” of a compound? (Because also things like efficacy, potency, etc. could be considered hallmarks of quality.)
Answer: the phrase was rewritten in another way, keeping the original meaning: “… are generally safer, effective and with less side-effects when compared …” as suggested.
L46: See my earlier comment about “highly conserved genome”. Molecular targets and physiological pathways and so on can be evolutionarily conserved, but a “conserved genome” is a bit odd (so I suggest rephrasing a bit).
Answer: the phrase was rewritten in another way, keeping the original meaning: “molecular targets and physiological pathways among vertebrates …”.
L48: It would be interesting if you could briefly summarize here what are the most commonly used phenotype-based screenings, if not with zebrafish.
Answer: A table was inserted in the manuscript to inform selected studies with larval zebrafish
L56-67: I agree that working with embryos is a way to accommodate legislation regarding the ethical usage of animals, and I also agree that extrapolations can be made from zebrafish embryos to other vertebrates (including humans). Still, this is a double-faced coin, which means that there are not only advantages but also limitations to be considered. One important limitation is that larvae have a very rudimentary behavioral repertoire compared to adult fish, and are in that sense less versatile. For instance, larvae do not engage in mating/spawning behavior or full-extent shoaling behavior, but these can nevertheless be very relevant traits. As another example, testing the bioactivity of compounds to treat ageing/senescence-related diseases requires the use of adult/senescent fish rather than embryos or larvae. To characterize the bioactivity and toxicity of compounds to a fuller extent, recent papers in fish behavioral research have advocated for the use of fish in later life-stages in addition to only embryos or larvae, and I fully agree with that development (see e.g. DOI: 10.1002/etc.4301 and DOI: 10.1016/j.chemosphere.2021.129697). I believe some discussion on this point should be added to the manuscript (it can be later on), to recognize that there are also limitations to embryo/larvae-based fish testing.
L63-67: Research with zebrafish early-life stages “obeys the concept of 3Rs” from the sense that they are not considered as laboratory animals (“Replacement”) and are therefore not subjected to any legislation regarding the ethical use of laboratory animals. It’s basically a legal backdoor. Building on my previous comment, I would add that tests with adult (zebra)fish can also be conform the legal ethical framework, by investing in “Reduction” and “Refinement”, as discussed in DOI: 10.1111/jfb.14512 and DOI: 10.1016/j.chemosphere.2021.129697 by the aforementioned author-group. In my opinion, adding such perspective is important because it gives a more nuanced and complete picture of the latest developments in the field of fish testing.
Answer: the follow text was inserted in the manuscript: “Chemical screenings rely on biological models, generally well-characterized and suitable to extrapolation to other vertebrates. Ideally, the use of models with established methods is required for reliable results. Zebrafish behavioral repertoire in larval stage is limited, since the nervous system is still in development, and several neurocircuits could be absent or underdeveloped. Thus, besides being an undoubtably suitable biomedical model, zebrafish larvae may not express certain complexes behavior. Behaviors like schooling, aggressive encounters and mating, only appears at the transition to adults, with these behaviors comprising social behavior domains, which become fully developed in adult fishes (Orger and Polavieja, 2017; Thoré et al., 2019)”.
L75-84: I fully agree, great section!
Answer: The authors appreciate your valuable opinion.
- Zebrafish Neurotransmitter Systems
L126-128: I suggest to tweak this sentence a bit, for instance like: “ Noradrenaline (NA) acts on the autonomic nervous system and controls cognition including learning and memory, as well as arousal and reward systems.”
Answer: the phrase has been adjusted as suggested.
L143: Please explain the abbreviation upon first encounter (please also check this for other abbreviations, I didn’t verify it all).
Answer: All the abbreviations were clarified.
L148-153: Yes, but see my above comments on the limitations of larvae tests
Answer:
- Neurological Functions and Behavior Models Towards Pre-Clinical Assays
L159-161: I agree, tests with zebrafish larvae are much more efficient compared to when you use older animals. But again, I believe that some discussion of its limitations is necessary to provide a balanced view.
Answer:
L179-187: Please specify if these statements are specifically for adults, larvae, or both.
Answer: The phrase “… in both larvae and adult …” was inserted in the text to clarify.
L192-194: I like that you provide an overview of which behavioral assays exist in zebrafish, great work! Are these the only established zebrafish larvae behavioral tests, or are there others (i.e. is this list exhaustive)?
Answer: A table was inserted in the manuscript to inform selected studies with larval zebrafish
L206-207: “since zebrafish responds in a fast manner by altering another behavior like exploration and boldness” This part of the sentence, and its connection with activity level is not so clear. Please rephrase to clarify.
Answer: the phrase was rewritten in another way, keeping the original meaning: “since zebrafish responds in a fast manner by altering locomotor parameters such distance moved and mean velocity, translating in behaviors like increased exploration.”
L210-213: This sentence needs some tweaking. Perhaps something like this? “They used the automated tracking device ZebraBoxTM to evaluate acute locomotor impairment in zebrafish larvae, as well as their escape response upon slightly disturbing the tail with a fine needle.”
Answer: the phrase was rewritten in another way, keeping the original meaning: to evaluate acute locomotor impairment in zebrafish larvae, as well as their escape response upon slightly disturbing the tail with a fine needle.”, as suggested.
L225: Please change “performs” to “perform”.
Answer: the word was altered as suggested.
L241-245: This explanation is a bit confusing, please clarify (i.e. increased anxiety is reflected by reduced activity and by spending less time in the open and illuminated area).
Answer: The follow phrase was inserted to compliment the rationale: “Thus, zebrafish larvae with decreased anxiety levels exhibit positive phototaxis. “
L245-248: I’m not sure if I get your point here, could you please rephrase to clarify?
Answer: the paragraph was rewritten to clarify: “Copmans et al. (2019) identified two known isoquinoline alkaloids TMC-120A and TMC-120B as novel antiseizure compounds, which were tested using the zebrafish larvae photomotor response (PMR) assay [96]. The PMR is a stereotypical behavior of 30–40 hpf zebrafish embryos, observed by embryo’s movements inside the chorion, and is triggered by high-intensity light [19,97].”
L264: I must admit that this paragraph confuses me a bit, primarily because I’m not entirely sure what you mean by “visual behavior”? Do you mean the movement of the eyes? How does one make these observations (i.e. which equipment/assays etc.)?
Answer: the paragraph was rewritten to clarify: “… in other words, alteration of locomotor parameters mediated by visual stimuli. The optomotor response (OMR) and optokinetic response (OKR) are innate visual responses in which individual moves the eye (OKR) and/or swims in the direction of an optic flow (OMR) and mimics a naturalistic behavior in larval zebrafish, aiming to stabilize the image around (Orger and Polavieja, 2017). Another visual behavior, the visuomotor response (VMR) aims to evaluate both locomotion ability and visual capacity of zebrafish larvae. In this test, zebrafish larvae are exposed to alternating light and dark periods, and individuals with functional visual system should respond showing specific swimming patterns (Orger and Polavieja, 2017). Zebrafish characteristic of …”.
- Concluding remarks
L287: Please change “decision-make processes” to “decision-making processes”. (Also applies for L308).
Answer: the expression was altered as suggested.
L289: Please change “apparatus” to “apparatuses” or “devices”.
Answer: the word was altered as suggested.
Figure 1: The figure is visually very attractive, great work! One point for improvement, however, is that the link between the displayed behaviors (purple box) and the text is not so clear. In the text, you discussed 4 assays, in the following order: 1. locomotor test in a Petri-plate or multi-well plate, 2. escape test upon disturbance, 3. visual motor response, and 4. light/dark preference test. The order in the figure, as well as the name of the tests, seems to be different. Please use the same order and name for clarity/consistency.
Answer:

Reviewer 2 Report
In the present review article entitled “Zebrafish larvae behavior models as a tool for drug screenings and pre-clinical trials: a review”, João Gabriel Santos ROSA and the Co-authors have literature surveyed to review the discovery of new therapeutic substances, drug development and research relies in trustful biological systems to obtain effective results. Moreover, this review summarizes the multi-targets strategies for treating complex diseases, and zebrafish emerged as an important model to biomedical research and drug screenings. Zebrafish clear correlation between neuro-anatomical and physiological features and behavior are very similar to that verified in mammals, enabling the construction of reliable and relevant experimental models for neurological disorders research. Zebrafish presents a highly conserved genome among vertebrates and a robust behavioral repertory. This review highlights important behavioral domains studied in zebrafish larvae and its neurotransmitter systems and summarizes currently used techniques to evaluate and quantify zebrafish larvae behavior in laboratory studies.
However, some major points need to be addressed, as mentioned below
Even though the authors describe the highlights of important behavioral domains studied in zebrafish larvae and its neurotransmitter systems and summarizes currently used techniques to evaluate and quantify zebrafish larvae behavior in laboratory studies, but the authors need to have a sub section for Zebrafish model for development phenotype which can cover other related phenotypes of zebrafish development to detect the efficiency of small molecules on developmental process or find out the teratogenic effect of available food products on development in this section.
The authors can add one or two figure particularly illustrating the role of zebrafish development to detect the efficiency of small molecules on developmental process or find out the teratogenic effect of available food products on development.
Some minor points also need to be addressed, as mentioned below,
The authors need to check the abbreviations, in first time usage they need to use the full word and abbreviation. For example “hpf “
The authors need to add a table to inform the collective studies which has used zebrafish as a model for their studies to explain the phenotype of the study model.
A concluding figure can be added, and it can illustrate the mechanisms of using zebrafish for the related studies. (the authors can use biorender to prepare figures for this manuscript)
There are few grammatical errors the authors need to check carefully to improve the manuscript quality.
The authors have shown the role of behavioral domains studied in zebrafish larvae and its neurotransmitter systems and summarizes currently used techniques to evaluate and quantify zebrafish larvae behavior in laboratory studies. The available research information seems to be sufficient, and the authors need to address the above comments. Taking together to all this issue I recommend minor revision of the manuscript.
Author Response
Manuscript ID: ijms-1754880
Type of manuscript: Review
Title: Zebrafish larvae behavior models as a tool for drug screenings and pre-clinical trials: a review
Authors: Joao Gabriel Santos Rosa, Carla Lima, Monica Lopes-Ferreira
First of all, we appreciate you taking the time out to share your comments with us. We value and respect your opinion. We agree with the need to make the text more appropriate for the Journal's audience, and we have already adjusted the manuscript, including additional changes as suggested.
Reviewer #2
The authors can add one or two figure particularly illustrating the role of zebrafish development to detect the efficiency of small molecules on developmental process or find out the teratogenic effect of available food products on development.
Answer: A figure was added to illustrate the effect of new drugs on developmental stages in zebrafish.
Some minor points also need to be addressed, as mentioned below,
The authors need to check the abbreviations, in first time usage they need to use the full word and abbreviation. For example, “hpf “
Answer: The abbreviations were revised.
The authors need to add a table to inform the collective studies which has used zebrafish as a model for their studies to explain the phenotype of the study model.
Answer: A table was inserted in the manuscript to inform selected studies with larval zebrafish.
A concluding figure can be added, and it can illustrate the mechanisms of using zebrafish for the related studies. (the authors can use biorender to prepare figures for this manuscript)
Answer: Two figures are part of the manuscript, illustrating the use of larval zebrafish on clinical trials.
There are few grammatical errors the authors need to check carefully to improve the manuscript quality.
Answer: The manuscript was reviewed to assure the proper use of language.
The authors have shown the role of behavioral domains studied in zebrafish larvae and its neurotransmitter systems and summarizes currently used techniques to evaluate and quantify zebrafish larvae behavior in laboratory studies. The available research information seems to be sufficient, and the authors need to address the above comments. Taking together to all this issue
I recommend minor revision of the manuscript.

Round 2
Reviewer 1 Report
The authors did a great effort to improve the manuscript. They amended the manuscript based on my comments and suggestions, added more nuance and included a table to illustrate commonly-studied behavioral endpoints in zebrafish larvae.
I only have some minor comments left, which I list below. I hope this will help to dot the i’s and cross the t’s. (Note that I didn’t include line-by-line comments, because there were no line numbers in the provided PDF-file.)
Great work, and best wishes!
- Last sentence of first paragraph: please provide a reference for this statement. (Reference to literature in these first two paragraphs is generally quite scarce, so it would be good if you could add some reference to literature still for some of these statements.)
- Last word of the second paragraph: note that "screens" should be "screenings". (I also saw this in other places, for instance in paragraph 3, so it would be good to fix that throughout the manuscript.)
- At the end of the first paragraph of section 3.1, I suggest to add some nuance. Altered locomotor activity (e.g. activity level, velocity) can, but does not necessarily have to, lead to changes in exploration behavior. Hence, I suggest to rephrase to something like: "since zebrafish responds in a fast manner by altering locomotor parameters such as distance moved and mean velocity, which could lead to changes in exploration"
- 6th paragraph of section 3.1: You say that non-anxious larvae in the light-dark preference test tend to increase exploration behavior and spend more time in the boundary zone (thigmotaxis). However, it should be the other way around: non-anxious (bold) larvae spend more time in the center of the tank, while anxious fish spend more time in the periphery of the tank (close to the walls), see e.g. DOI: 10.1111/jfb.14512. Please also add a reference to literature for this latter statement.
- First sentence of section 3.2: This is now more clear to me, great!
- Figures: It seems like the figures are not included in the revised version of the manuscript, so I couldn't review them anymore.
- Table: Cool table, it gives a good idea of the kind of endpoints that are often assessed. I suggest to switch the order of the columns though: 1. Behavioral test, 2. Endpoints, 3. Reference. I would delete the DOI column, and instead include the references in the reference list with full bibliographic details.
- Reference list: please check if all the sources are correctly referenced. For instance, reference n°79 ("Noldus No Title") seems to be incorrectly referenced?
Author Response
Manuscript ID: ijms-1754880 R2
Type of manuscript: Review
Title: Zebrafish larvae behavior models as a tool for drug screenings and pre-clinical trials: a review
Authors: Joao Gabriel Santos Rosa, Carla Lima, Monica Lopes-Ferreira
The authors did a great effort to improve the manuscript. They amended the manuscript based on my comments and suggestions, added more nuance and included a table to illustrate commonly studied behavioral endpoints in zebrafish larvae.
I only have some minor comments left, which I list below. I hope this will help to dot the i’s and cross the t’s. (Note that I didn’t include line-by-line comments, because there were no line numbers in the provided PDF-file.)
Great work, and best wishes!
- Last sentence of first paragraph: please provide a reference for this statement. (Reference to literature in these first two paragraphs is generally quite scarce, so it would be good if you could add some reference to literature still for some of these statements.)
Answer: The sentence was slightly changed, preserving the original meaning, and it was referenced.
- Last word of the second paragraph: note that "screens" should be "screenings". (I also saw this in other places, for instance in paragraph 3, so it would be good to fix that throughout the manuscript.)
Answer: The word spelling was corrected, as well as the other words in the text.
- At the end of the first paragraph of section 3.1, I suggest to add some nuance. Altered locomotor activity (e.g. activity level, velocity) can, but does not necessarily have to, lead to changes in exploration behavior. Hence, I suggest to rephrase to something like: "since zebrafish responds in a fast manner by altering locomotor parameters such as distance moved and mean velocity, which could lead to changes in exploration"
Answer: The phrase was corrected as suggested.
- 6th paragraph of section 3.1: You say that non-anxious larvae in the light-dark preference test tend to increase exploration behavior and spend more time in the boundary zone (thigmotaxis). However, it should be the other way around: non-anxious (bold) larvae spend more time in the center of the tank, while anxious fish spend more time in the periphery of the tank (close to the walls), see e.g. DOI: 10.1111/jfb.14512. Please also add a reference to literature for this latter statement.
Answer: The conceptual mistake was corrected and the reference was added.
- First sentence of section 3.2: This is now more clear to me, great!
Answer: The authors thank you for your support.
- Figures: It seems like the figures are not included in the revised version of the manuscript, so I couldn't review them anymore.
- Table: Cool table, it gives a good idea of the kind of endpoints that are often assessed. I suggest to switch the order of the columns though: 1. Behavioral test, 2. Endpoints, 3. Reference. I would delete the DOI column, and instead include the references in the reference list with full bibliographic details.
Answer: The table was altered as suggested.
- Reference list: please check if all the sources are correctly referenced. For instance, reference n°79 ("Noldus No Title") seems to be incorrectly referenced?
Answer: This and other references have been corrected and revised.